# Refining the diagnostic approach to latent tuberculosis Infection with Quantiferon gold plus: A retrospective analysis of borderline results

Alba Ruedas-López[1,2☯], Juan María Herrero-Martínez[3☯], Alhena Reyes[1], Beatriz González-Blanco[1], Paula López-Roa[1]*

1 Clinical Microbiology and Parasitology Department, Hospital Universitario 12 de Octubre, Madrid, Spain,
2 Department of Preventive Medicine, Public Health and Microbiology, School of Medicine, Universidad Autónoma de Madrid, Madrid, Spain, 3 Internal Medicine Department, Hospital Universitario 12 de Octubre, Madrid, Spain

☯ These authors contributed equally to this work.
* plroa@salud.madrid.org

## Abstract

The Quantiferon Gold Plus (QFT) test, a widely used interferon-γ release assay (IGRA), diagnoses latent tuberculosis infection (LTBI) with a positivity threshold of ≥0.35 IU/mL. Results near this cut-off can be challenging to interpret due to variability from immunological, pre-analytical, and technical factors, prompting recommendations for a borderline range to refine diagnosis and reduce overtreatment. This retrospective study analyzed QFT results from 9,944 patients (2019–2023), establishing ranges: <0.2 IU/mL as negative, 0.2–0.35 IU/mL as borderline negative, 0.35–0.7 IU/mL as borderline positive, and >0.7 IU/mL as positive. Borderline results occurred in 7.6% of patients, particularly in those born in Africa or South America, and in older individuals. Of 64 patients retested, 60.9% reverted to negative, while 17.1% of borderline negatives later converted to positive or borderline positive. Notably, no active TB cases emerged among those who reverted to negative on repeat testing. These findings emphasize the need for cautious interpretation of borderline QFT results, as their link to active TB progression differs from clear results. The study supports repeat testing of borderline cases to enhance LTBI diagnostic accuracy and inform treatment decisions.

## Introduction

The Quantiferon Gold Plus (QFT) test is one of the two commercially available interferon-γ release assays (IGRAs) used for diagnosing latent tuberculosis infection (LTBI). Compared to its predecessor, the QuantiFERON-TB Gold In-Tube (QFT-GIT), the QFT-Plus has demonstrated improved sensitivity [1], while maintaining

**Data availability statement:** The data that support the findings of this study are openly available in Figshare. The data can be accessed at Figshare using the following DOI link: https://doi.org/10.6084/m9.figshare.28659293.v1.

**Funding:** This study has been funded by Instituto de Salud Carlos III (ISCIII) through the project 'PI21/01738' and co-funded by the European Union.

**Competing interests:** The authors have declared that no competing interests exist.

comparable or, in some studies, improved specificity [2,3]. Additionally, QFT-Plus may offer greater sensitivity for predicting active TB [4]. However, IGRA tests are more expensive than the tuberculin skin test (TST), which may remain a cost-effective option in some high TB burden countries. Factors such as cost-effectiveness and local epidemiological conditions, including TB prevalence and resource availability, are critical when choosing between these tests [3]. In our clinical practice, we have adopted the QFT-Plus as the primary diagnostic tool for LTBI. As per the manufacturer's guidelines [5], a QFT result is classified as positive if it is ≥ 0.35 IU/mL [6]. However, interpreting IGRA results can be challenging due to issues such as indeterminate and equivocal results, including conversions (from negative to positive) or reversions (from positive to negative) during repeat testing [7].

Longitudinal studies have observed variability in IGRA results over time, particularly when the outcomes are near the threshold. These studies report a significant occurrence of conversions and reversions in borderline results [8,9]. This fluctuation is believed to arise from pre-analytical, technical, or patient-related factors [7].

Given the high frequency of conversions and reversions around the 0.35 IU/mL threshold, some researchers have recommended establishing a borderline or equivocal range [10]. This approach aims to reduce unnecessary treatment for individuals with unstable positive results while ensuring appropriate treatment for those close to the threshold who may convert to positive on retesting [11]. Suggested borderline ranges vary across studies, with some recommending 0.20–0.70 IU/mL and others extending up to 1.0 IU/mL [8,12–14].

In addition to borderline results, QFT outcomes may be uninterpretable if the positive (mitogen) or negative control tubes fail to provide valid responses. Such results are labeled as indeterminate, often necessitating a repeat test. Under routine conditions, the rate of indeterminate outcomes has been reported to range between 3.4% and 10.2% [15,16].

In our setting, limited data exist on the frequency of borderline IGRA results and associated risk factors under routine clinical conditions. The objective of this study was to determine the proportion of QFT tests that yield neither clear positive nor negative outcomes in clinical practice and to evaluate the results of repeating equivocal tests.

## Methods

A retrospective review of patients with QFT results between 2019 and 2023 was conducted, including demographic data such as age, gender, and country of birth. These variables have been associated, along with borderline test results, with both the likelihood of conversions and reversions of Interferon γ Release Assays [7]. The study was conducted in a routine clinical setting at a tertiary hospital in Madrid, focusing on patients of all age groups who underwent QFT testing. Children were not excluded, as previous research suggests that the QFT assay is accurate in pediatric practice, with good sensitivity [17]. The hospital is a 1,300-bed acute teaching hospital that serves as a referral centre for a population of 462,000 [18], of whom 33.6% are foreign-born [19]. The population is comprised of 69% individuals between the ages

of 14 and 64 years, 11% between 65 and 79 years, and 6% over 80 years. The incidence of tuberculosis in the population of the three health districts attached to the hospital is among the highest in the Community of Madrid, ranging from almost 19 cases per 100,000 inhabitants in the districts of Usera or Villaverde, to 14.8 per 100,000 in Carabanchel, compared to the Community average of 8.59 per 100,000 [20].

Data for research purposes were accessed from January 2024 to May 2024. Patients were tested based on clinical indications, such as suspected latent or active tuberculosis, or for screening purposes. Repeat testing was carried out when deemed necessary by the attending physicians, in consultation with microbiologists. This occurred in cases where there were inconsistencies with expected results, a high clinical suspicion of TB, or indeterminate results.

The assay was performed using chemiluminescence on the Liaison XL platform (DiaSorin S.p.A., Saluggia, Italy), with results interpreted according to the recommended cut-off point (0.35 IU/mL).

We aimed to assess the proportion of QFT tests yielding equivocal results and to evaluate the outcomes of retesting these cases by establishing a defined borderline range. The results were categorized as follows: negative result: < 0.2 IU/mL; borderline negative: ≥ 0.2 and <0.35 IU/mL; borderline positive: 0.35–0.7 IU/mL; positive: > 0.7 IU/mL.

We employed a logistic regression model to assess the association between demographic variables and QFT results.

## Ethical statement

This study was conducted in accordance with the ethical principles outlined in the Declaration of Helsinki and was approved by the Institutional Ethics Committee of Hospital 12 de Octubre (reference number: CEIm 23/569), ensuring compliance with international ethical standards. All procedures adhered to the hospital's biosafety regulations and were approved by the Local Institutional Review Board. Given the retrospective nature of the study, all data were fully anonymized prior to access. Additionally, due to the retrospective design, the Ethics Committee waived the requirement for written informed consent, as no identifiable patient information was included in the analysis.

## Results

We identified a total of 9944 individuals who underwent at least one QFT test, with 5020 (50.5%) being male. The study population comprised individuals aged from 0.05 to 98.52 years, with a mean age of 51.64 years (SD = 19.24). Focusing on younger subgroups, individuals under 18 years of age accounted for 4.7% (466 patients), while those under 5 years represented only 1.2% (122 patients) of the total population. The average age was 51.7 years (SD = 19.4) for men and 51.6 years (SD = 19.0) for women. Information about the country of birth was available for 9040 participants (90.9%), among whom 2684 (29.7%) were born outside the country.

Using the standard cut-off of 0.35 IU/mL, 1805 (18.1%) tests were classified as positive, 7872 (79.2%) as negative, and 267 (2.7%) as indeterminate. When introducing a borderline range, 7506 (75.5%) results fell into the negative category, 1413 (14.2%) were positive, 392 (3.9%) were classified as borderline positive, and 366 (3.7%) as borderline negative. In total, 7.6% (758/9944) of those tested had values within the borderline range (Fig 1).

Compared to individuals born in Europe, those born in Africa and South America were significantly more likely to have borderline results. A slight increase in age was also associated with a higher probability of borderline outcomes, while no notable differences were found between males and females. Details are provided in Table 1.

Among 64 patients with an initial borderline result who underwent repeat testing, the outcomes were as follows: 3 (4.7%) were indeterminate, 39 (60.9%) negative, 9 (14.1%) borderline negative, 6 (9.4%) borderline positive, and 7 (10.9%) positive. For patients initially classified as borderline negative, 17.1% converted to positive (9.8%) or borderline positive (7.3%) on retesting. In contrast, among those initially categorized as borderline positive, 73.9% reverted to negative (52.2%) or borderline negative (21.7%) upon retesting. Categorical outcomes of repeat testing are detailed in Table 2 and both categorical and demographic data, including continent and mean age, are presented in S1 Table.

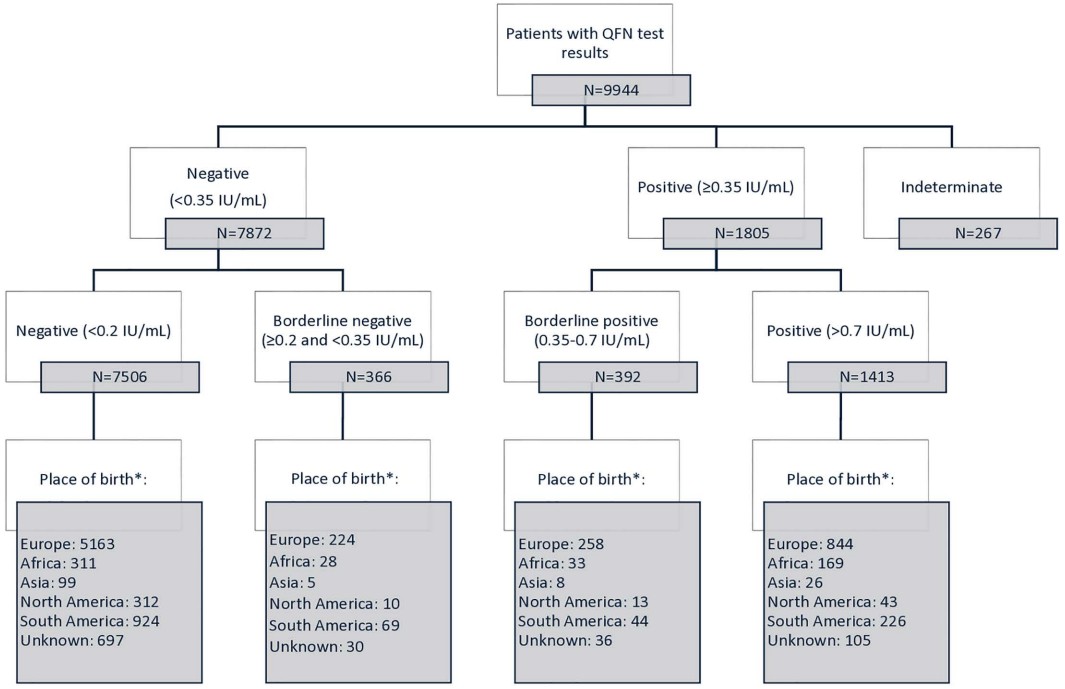

**Fig 1. Distribution of test results by cut-off ranges.**

**Table 1. Analysis of variables associated with a borderline result\*.**

| Variable | | Results\*\* N (%) | | | | Adjusted OR (all borderline vs. all definitive positive or negative results) | |
|---|---|---|---|---|---|---|---|
| | | Negative | BL-neg. | BL-pos. | Positive | OR$_a$ (95% CI) | p |
| Place of Birth (OR, relative to Europe) | Europe | 5163 (79.6) | 224 (3.5) | 258 (4.0) | 844 (13.0) | – | – |
| | Africa | 311 (57.5) | 28 (5.2) | 33 (6.1) | 169 (31.2) | 2.05 (1.54-2.74) | **<0.001** |
| | Asia | 99 (71.7) | 5 (3.6) | 8 (5.8) | 26 (18.8) | 1.62 (0.90-2.91) | 0.1 |
| | North America | 312 (82.5) | 10 (2.7) | 13 (3.4) | 43 (11.4) | 1.12 (0.72-1.73) | 0.6 |
| | South America | 924 (73.2) | 69 (5.5) | 44 (3.5) | 226 (17.9) | 1.59 (1.27-1.98) | **<0.001** |
| | Unknown | 697 (80.3) | 30 (3.5) | 36 (4.2) | 105 (12.1) | 1.26 (0.96-1.66) | 0.1 |
| Female | | 3899 (80.9) | 177 (3.7) | 183 (3.8) | 560 (11.6) | – | – |
| Male | | 3607 (74.3) | 189 (3.9) | 209 (4.3) | 853 (17.6) | 1.11 (0.95-1.28) | 0.2 |
| Age (years, median (IQR)) | | 50.6 (36.8-62.7) | 56.6 (44.0-71.8) | 59.4 (47.4-73.6) | 58.5 (46.5-71.1) | 1.02(1.02-1.03) | **<0.001** |

Note. "BL-pos.": borderline positive results; "BL-neg.": borderline negative results.

\* Patients with indeterminate results were excluded from the analysis. N = 9677.

\*\* Negative: < 0.2 IU/mL; borderline negative: ≥ 0.2 and < 0.35 IU/mL; borderline positive: 0.35–0.7 IU/mL; positive: > 0.7 IU/mL.

During the study period, 90 individuals were diagnosed with active tuberculosis (TB) among those included in our study. QFT and culture results for these cases are presented in Table 3.

Excluding those with indeterminate or negative results, there was a significantly higher proportion of patients developing active TB in those with positive results >0.7 IU/mL (4.8%; 68/1413) compared to individuals with values within the borderline range of 0.20–0.7 IU/mL (1.1%; 8/757, p < 0.001).

**Table 2. Categorical distribution of follow-up QFT results when retesting those with initial results in the borderline range (0.20–0.70 IU/ml).**

| Initial result | Total | Retested N (%) | Results of follow-up QFT test* N (%) | | | | | Months until retesting | |
|---|---|---|---|---|---|---|---|---|---|
| | | | Ind. | Negative | BL negative | BL positive | Positive | Mean (SD) | Range |
| **BL negative** | 366 | 41 (11.2) | 3 (7.3) | 27 (65.9) | 4 (9.8) | 3 (7.3) | 4 (9.8) | 15.1 (10.7) | 0.1 - 49.2 |
| **BL positive** | 392 | 23 (5.9) | 0 (0) | 12 (52.2) | 5 (21.7) | 3 (13.0) | 3 (13.0) | 10.94 (9.4) | 0.03 - 29.9 |
| **All borderline** | 758 | 64 (8.4) | 3 (4.7) | 39 (60.9) | 9 (14.1) | 6 (9.4) | 7 (10.9) | 13.6 (10.4) | 0.03 - 49.2 |

Note. **BL**: borderline, **Ind.**: Indeterminate. * Negative: < 0.2 IU/mL; borderline negative: ≥ 0.2 and <0.35 IU/mL; borderline positive: 0.35–0.7 IU/mL; positive: > 0.7 IU/mL.

**Table 3. Mycobacterial culture according to Quantiferon test result.**

| QFT results | Positive culture results | | |
|---|---|---|---|
| | Co-prevalent (<3 months) | Incident (>3 months) | Total |
| Indeterminate | 4 (5.1%) | 0 (0.0%) | 4 (4.4%) |
| Negative | 9 (11.4%) | 1 (9.1%) | 10 (11.1%) |
| BL-Neg | 3 (3.8%) | 2 (18.2%) | 5 (5.6%) |
| BL-Pos | 2 (2.5%) | 1 (9.1%) | 3 (3.3%) |
| Positive | 61 (77.2%) | 7 (63.6%) | 68 (75.6%) |
| Total | 79 (87.8%) | 11 (12.2%) | 90 (100%) |

Note. QFT, Quantiferon; BL-Neg, borderline negative; BL-Pos, borderline positive.

Among individuals with initial QFT results in the 0.20–0.7 IU/mL range who underwent retesting, 3.1% (2/64) progressed to active TB. One patient initially classified as borderline negative and another as borderline positive later tested positive during follow-up. For those with borderline positive results (0.35–0.7 IU/mL), reversions to negative were observed in 52.2% (12/23), and no cases of active TB were detected among individuals who reverted.

## Discussion

In this evaluation of borderline QFT results obtained under routine clinical conditions in a low-endemicity setting involving 9,944 individuals, nearly 8% of tests yielded results within the borderline range. Interestingly, among those retested after an initial borderline-range result, 60.9% showed convincingly negative results (<0.20 IU/ml). Additionally, 73.9% of individuals initially classified as borderline positive reverted to either negative or borderline negative, with no new TB cases identified in this group. However, 17.1% of those initially categorized as borderline negative progressed to borderline positive or positive results, suggesting a potential underestimation of latent TB risk. These findings highlight the challenge of interpreting single borderline results, which can result in both unnecessary treatments and missed LTBI cases.

Variability in QFT results may stem from immunological, pre-analytical, and technical factors [21]. This variability has been extensively studied in the context of serial testing among healthcare workers (HCWs). For example, studies in large HCW populations in the USA showed that 65–75% of individuals who converted from negative to positive reverted to negative upon repeat testing [22,23]. Such fluctuations have prompted recommendations to introduce a borderline range to aid interpretation. A German study, applying a borderline zone of 0.2–0.7 IU/mL, demonstrated a reduction in reversion rates from 37% to 19% [8]. Similarly, 43% of HCWs in Sweden with borderline QFT results (0.30–0.99 IU/mL) showed reversion upon retesting [14]. However, beyond HCW-focused studies, limited data exist on the prevalence of borderline results and their associated risk factors in routine settings. Our analysis revealed that approximately 8% of QFT

tests resulted in borderline findings, with the progression risk to active TB likely differing from clearly positive or negative results.

Consistent with previous studies [24–26], our results suggest that borderline QFT values, particularly those that revert to negative, may reflect false-positive elevations rather than genuine immune responses to TB antigens. This hypothesis is supported by the absence of active TB cases in individuals whose borderline results reverted to negative.

Our multivariate analysis identified that individuals born in Africa or South America were significantly more likely to obtain borderline results. Age was also found to slightly increase the likelihood of borderline results, while no significant sex-based differences were observed. The higher percentages of borderline QFT results observed in Africa and South America may be linked to the higher prevalence of TB in these regions. While this is a plausible explanation, it remains speculative, as TB prevalence was not included in our regression model. It is also important to note that certain species of non-tuberculous mycobacteria (NTM), such as *M. kansasii*, *M. marinum*, and *M. szulgai*, can cause positive QFT results [5]. However, due to the limited availability of comprehensive data on the prevalence of these NTMs across different continents, it is not possible to determine whether this factor influences the observed differences in borderline percentages. Nonetheless, regardless of the underlying cause, there appears to be added value or greater indication for retesting borderline results in these populations.

Interpreting diagnostic test results requires careful consideration of the clinical and epidemiological context. The risk of developing active TB varies among individuals with positive IGRA reactions. Studies have demonstrated a correlation between IGRA result magnitude and the incidence of active TB, including cases within the borderline range, suggesting a dose-response relationship [27]. However, in regions with low TB prevalence, individuals presenting with borderline QFT results often experience reversion to negative upon retesting. A study involving 58,539 subjects in a low-endemicity country found that among those with initial borderline results who were retested, 38% reverted to negative, with no cases of incident active TB reported within two years. This finding suggests that a significant proportion of initial borderline results may represent false positives [26]. Consequently, the risk of missing a true LTBI in such cases is relatively low, and avoiding unnecessary treatment minimizes the potential for adverse effects and disruptions to daily life.

Conversely, in populations at higher risk for TB progression, such as immunosuppressed individuals, the emphasis shifts toward maximizing test sensitivity, even at the expense of specificity. Failure to diagnose and treat LTBI in these patients can lead to severe outcomes, including the development of active TB disease. Therefore, integrating other risk factors—such as recent exposure to active TB cases, underlying comorbidities, and the degree of immunosuppression—along with underlying determinants such as access to healthcare, socioeconomic status, and level of vulnerability is essential in guiding clinical decisions [28].

Contrasting evidence suggests that most borderline QFT results reflect true antigen-specific responses rather than random variability, and that the presence of various non-IGRA risk factors and parameters of TB infection showed a gradient along increasing quantitative QFT results [29,30]. This study provided interesting data on QFT results, but it is important to note that the study was conducted within the context of a contact study, and not in other settings, where the interpretation of the results and the risk of progression to active TB may vary significantly. We agree with Wikell et al. [26] that the inherent variability of the technique should be given due consideration, and that further study is required to ascertain the clinical significance of borderline results in terms of the development of active TB in a range of clinical settings.

Furthermore, given the proportion of borderline QFT results observed in our study and the low incidence of TB progression within this subgroup, one could argue that clinical follow-up alone may suffice, with treatment initiated as necessary. However, refining diagnostic classification strategies would enhance the efficient allocation of healthcare resources and could also reduce the stigma associated with a positive diagnosis. Targeting preventive treatment to individuals most likely to benefit not only optimizes cost-effectiveness but also reduces unnecessary interventions in those at low risk for disease progression.

From a public health perspective, any modifications to TB screening protocols must consider the availability of resources—both material and personnel—to ensure appropriate patient follow-up. Implementation strategies should be tailored to local epidemiological conditions and healthcare infrastructure to maximize feasibility and impact.

Given the significant morbidity and mortality associated with TB, further research is essential to determine which individuals with IGRA conversion or reversion are most likely to benefit from TB preventive treatment, and more studies are warranted across diverse geographical regions and ethnic populations. A deeper understanding of the clinical significance of borderline results for specific patient populations will be crucial in refining LTBI management strategies and improving patient outcomes.

This study has several limitations. First, it relied on retrospectively collected clinical data, and not all equivocal or indeterminate results were retested. Second, data on the reasons for testing and whether LTBI treatment was initiated were unavailable. Third, the observed relationship between QFT results and active TB development should be interpreted cautiously due to the retrospective design and the small number of borderline cases progressing to active TB. Additionally, some patients may have been diagnosed with TB at other institutions, potentially affecting the results.

In conclusion, approximately 8% of patients had borderline QFT results, with these findings being more prevalent among individuals born in South America or Africa, as well as older individuals. Patients with borderline positive results showed a high reversion rate (about 50%), and no cases of TB were identified among those who reverted. These findings emphasize the importance of retesting and cautious interpretation of borderline QFT results in clinical practice. While repeat testing offers valuable insights, it may not always be feasible, and further studies are needed to assess the long-term clinical outcomes of borderline QFT results.

## Supporting information

**S1 Table.  Mean age and categorical distribution of follow-up QFT results when retesting those with initial results in the borderline range (0.20–0.70 IU/ml), by continent.**
(DOCX)

## Author contributions

**Data curation:** Alba Ruedas-Lopez, Alhena Reyes, Beatriz Gonzalez-Blanco.

**Formal analysis:** Alba Ruedas-Lopez, Alhena Reyes, Beatriz Gonzalez-Blanco.

**Investigation:** Juan Maria Herrero-Martinez, Paula López-Roa.

**Methodology:** Juan Maria Herrero-Martinez, Paula López-Roa.

**Supervision:** Paula López-Roa.

**Writing – original draft:** Alba Ruedas-Lopez, Juan Maria Herrero-Martinez.

**Writing – review & editing:** Paula López-Roa.

## Acknowledgments

This study has been funded by Instituto de Salud Carlos III (ISCIII) through the project "PI21/01738" and co-funded by the European Union.

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
