## [Decision Letter · Decision Letter 0]

12 Feb 2025

PONE-D-24-52042Refining the Diagnostic Approach to Latent Tuberculosis Infection with Quantiferon Gold Plus: A Retrospective Analysis of Borderline ResultsPLOS ONE

Dear Dr. Lopez Roa,

Thank you for submitting your manuscript to PLOS ONE. After careful consideration, we feel that it has merit but does not fully meet PLOS ONE’s publication criteria as it currently stands. Therefore, we invite you to submit a revised version of the manuscript that addresses the points raised during the review process.

We look forward to receiving your revised manuscript.

Kind regards,

Xiangwei Li

Academic Editor

PLOS ONE

Reviewers' comments:

Reviewer's Responses to Questions

**Comments to the Author**

1. Is the manuscript technically sound, and do the data support the conclusions?

Reviewer #1: Yes

Reviewer #2: Yes

Reviewer #3: Yes

Reviewer #4: Yes

Reviewer #5: Yes

Reviewer #6: Partly

2. Has the statistical analysis been performed appropriately and rigorously? 

Reviewer #1: Yes

Reviewer #2: Yes

Reviewer #3: Yes

Reviewer #4: Yes

Reviewer #5: Yes

Reviewer #6: Yes

3. Have the authors made all data underlying the findings in their manuscript fully available?

Reviewer #1: Yes

Reviewer #2: Yes

Reviewer #3: Yes

Reviewer #4: No

Reviewer #5: Yes

Reviewer #6: Yes

4. Is the manuscript presented in an intelligible fashion and written in standard English?

Reviewer #1: Yes

Reviewer #2: Yes

Reviewer #3: Yes

Reviewer #4: Yes

Reviewer #5: Yes

Reviewer #6: Yes

5. Review Comments to the Author

Reviewer #1: Congratulations for submitting this manuscript and for completing this study as part of the body of knowledge in the field of TB diagnosis.

Generally, the paper is clear and straightforward in addressing the research objective. In the methods section, you may specify the general objective you mentioned in the introduction. For instance, you could have explained more why did you choose the demographic variables and further describe what could be its implications when you include it in your study as far as QFT is concerned. In the resuls section, please be consistent with the use of proper formatting of tables based on the guidelines of the PLOS. Another suggestion would be to include contrasting studies (aside from consistent ones), whenever available. Additional references in the discussion section further improves the credibility of the findings. Lastly, you have a good conclusion.

May you keep on working on this kind of studies that would possibly inform future policy. God bless.

Reviewer #2: This retrospective review study aims to assess the (1) clinical characteristics of those with borderline tests and (2) the clinical applicability of borderline QFT-gold test results by analyzing the results of repeat testing (ie conversions/reversions) for the 7-8% of patients with borderline results on the initial test. Given the number of patients in the borderline category to reverted or converted, this study highlights the challenge of interpreting single borderline results and argues for repeat testing over time. This study did demonstrate that the risk of progressing to active disease in the borderline group (~5%) was different from that of both the negative (~0.1%) and positive group (~5%), though it should be noted that this study is retrospective in nature and sample numbers are small in the borderline group.

1. Did this study include children? It was not clear from the Methods section and given the emphasis on age as a correlate of borderline QFT results, I would recommend being explicit about the age range included in the study, especially if those <5yo are included.

2. Only 64/758 borderline patients were retested, but the clinical characteristics of this subset are not included in the study nor is the re-testing interval. While not all data may be available, it would be helpful to know if these patients had risk factors or other clinical characteristics different from the overall cohort that prompted the closer follow-up (for example, heavier exposure to TB, country of origin, symptoms of active TB, etc) compared to those that were not re-tested? Additionally, is it confirmed whether these 64 patients are treatment-naive given treatment data was not included as part of the study?

3. While re-testing of borderline QFT results may provide additional diagnostic clarity, retesting ~8% of patients is an additional health care cost and risks losing patients to follow-up, some of whom would go on to develop active TB, but has the advantage of potentially avoiding unnecessary LTBI treatment in a subset of individuals. It may be helpful to include in the discussion potential positive and negative consequences of initiating this intervention.

4. It would also be helpful to include in the discussion the effects of serial testing on public health in both low endemicity and high endemicity settings.

Reviewer #3: The manuscript entitled "Refining the Diagnostic Approach to Latent Tuberculosis Infection with Quantiferon

Gold Plus: A Retrospective Analysis of Borderline Results" needs these revisions

-Please check keywords according to MESH browser; delete QuantiFERON-TB Gold Plus and borderline results from keywords

-Please write about the age range of patients

-What are the rules for repeat testing, what is the minimum time interval for repeat testing

Reviewer #4: The manuscript is well written and highlighted the important subject of diagnosing LTBI, and more importantly to the borderline outcomes interrupting the results which alters in the retesting. Applause to the authors for extracting and analysing such data to develop this manuscript.

There are a couple of minor comments/suggestions to this.

Methods:

Line 75 - 86: this section could explore further the setup of health facility which the study was conducted, such as TB facility or any other, how patients were selected to test, QTF testing criteria, criteria for retesting of borderline results, and treatment and follow up or further care after LTBI diagnosis.

The demographic variables are only age, gender and country of birth. Are there any other variables included in the regression model such as previous TB history, HIV coinfection, or other comorbidities? It will be interesting to see the relation with such variables. Or are they not available as a limitation of the retrospective study.

Reslts:

Line 188 - 124: this is an interesting outcome after repeated testing. What is the time interval between initial testing vs repeated testing?

Reviewer #5: There is a very small percentage of patients who fall within the borderline range . Of these your classification improves the predictability of the classification. However only 1.1% of the cases develop TB during followup. It may not be too difficult to just follow-up those who fall in the border line range. So what is the added value of your improved Classification

You have said that the Percentages are more in the countries like Africa? Why is this so? Would the prevalence of NTM have something to do with it? This point could be addressed more in the discussion

Reviewer #6: In the introduction, it would be beneficial for the authors to highlight any existing studies that have evaluated and compared the results between QFT-Plus and QFT-GIT. Discussing any observed differences, such as costs and immune responses, would provide valuable insights, and including relevant references would enhance the credibility of the review.

In the materials and methods section, it is important to clearly indicate whether the patients are immunocompromised, part of a frail population, or have comorbidities. This information is essential, as these factors can significantly influence the interpretation of QFT results.

Furthermore, it is critical to emphasize that clinicians are not just interested in laboratory evaluation ranges, but in how these ranges are defined by population stratification, particularly by age. For instance, in Europe, the age demographic of the Caucasian population may differ. More importantly, the patient's overall health status plays a pivotal role in interpreting laboratory data. A holistic approach to clinical management should involve evaluating the population based on race and understanding the type of immune response, factoring in essential epidemiological and clinical variables such as cancer, autoimmune disorders, protein intake, BMI, coinfections, and access to healthcare, which can differ across countries.

6. PLOS authors have the option to publish the peer review history of their article (what does this mean? ). If published, this will include your full peer review and any attached files.

**Do you want your identity to be public for this peer review?** For information about this choice, including consent withdrawal, please see our Privacy Policy .

Reviewer #1: **Yes: ** Marlon L. Bayot

Reviewer #2: No

Reviewer #3: No

Reviewer #4: No

Reviewer #5: No

Reviewer #6: **Yes: ** none

---

## [Author Response · Author response to Decision Letter 1]

25 Mar 2025

Reviewer #1:

Congratulations for submitting this manuscript and for completing this study as part of the body of knowledge in the field of TB diagnosis. Generally, the paper is clear and straightforward in addressing the research objective.

Reviewer comment 1: In the methods section, you may specify the general objective you mentioned in the introduction.

Response: Thank you for your suggestion. We have addressed this by adding the following sentence to the Methods section “We aimed to assess the proportion of QFT tests yielding equivocal results and to evaluate the outcomes of retesting these cases by establishing a defined borderline range”

Reviewer comment 2: For instance, you could have explained more why did you choose the demographic variables and further describe what could be its implications when you include it in your study as far as QFT is concerned.

Response: Thank you for your comment. We agree that clarifying the importance of the selected demographic variables is relevant. To address this, we have added the following sentence to the Methods section: “These variables have been associated, along with borderline test results, with both the likelihood of conversions and reversions of Interferon γ Release Assays [1].”

Reviewer comment 3: In the resuls section, please be consistent with the use of proper formatting of tables based on the guidelines of the PLOS.

Response: Thank you for your feedback. We have reviewed and updated the formatting of the tables in the Results section to ensure they are consistent with the PLOS guidelines.

Reviewer comment 4: Another suggestion would be to include contrasting studies (aside from consistent ones), whenever available. Additional references in the discussion section further improves the credibility of the findings. Lastly, you have a good conclusion. May you keep on working on this kind of studies that would possibly inform future policy. God bless.

Response: Thank you for your valuable suggestion. We have include contrasting studies and expanded the discussion section with additional references. This has improved the contextualisation and overall understanding of the article.

“Interpreting diagnostic test results requires careful consideration of the clinical and epidemiological context. The risk of developing active TB varies among individuals with positive IGRA reactions. Studies have demonstrated a correlation between IGRA result magnitude and the incidence of active TB, including cases within the borderline range, suggesting a dose-response relationship [2]. However, in regions with low TB prevalence, individuals presenting with borderline QFT results often experience reversion to negative upon retesting. A study involving 58,539 subjects in a low-endemicity country found that among those with initial borderline results who were retested, 38% reverted to negative, with no cases of incident active TB reported within two years. This finding suggests that a significant proportion of initial borderline results may represent false positives [3]. Consequently, the risk of missing a true LTBI in such cases is relatively low, and avoiding unnecessary treatment minimizes the potential for adverse effects and disruptions to daily life.

Conversely, in populations at higher risk for TB progression, such as immunosuppressed individuals, the emphasis shifts toward maximizing test sensitivity, even at the expense of specificity. Failure to diagnose and treat LTBI in these patients can lead to severe outcomes, including the development of active TB disease. Therefore, integrating other risk factors—such as recent exposure to active TB cases, underlying comorbidities, and the degree of immunosuppression—along with underlying determinants such as access to healthcare, socioeconomic status, and level of vulnerability is essential in guiding clinical decisions [4].

Contrasting evidence suggests that most borderline QFT results reflect true antigen-specific responses rather than random variability, and that the presence of various non-IGRA risk factors and parameters of TB infection showed a gradient along increasing quantitative QFT results [5, 6]. This study provided interesting data on QFT results, but it is important to note that the study was conducted within the context of a contact study, and not in other settings, where the interpretation of the results and the risk of progression to active TB may vary significantly. We agree with Wikell et al. [3] that the inherent variability of the technique should be given due consideration, and that further study is required to ascertain the clinical significance of borderline results in terms of the development of active TB in a range of clinical settings.”

Reviewer #2

This retrospective review study aims to assess the (1) clinical characteristics of those with borderline tests and (2) the clinical applicability of borderline QFT-gold test results by analyzing the results of repeat testing (ie conversions/reversions) for the 7-8% of patients with borderline results on the initial test. Given the number of patients in the borderline category to reverted or converted, this study highlights the challenge of interpreting single borderline results and argues for repeat testing over time. This study did demonstrate that the risk of progressing to active disease in the borderline group (~5%) was different from that of both the negative (~0.1%) and positive group (~5%), though it should be noted that this study is retrospective in nature and sample numbers are small in the borderline group.

Reviewer comment 1. Did this study include children? It was not clear from the Methods section and given the emphasis on age as a correlate of borderline QFT results, I would recommend being explicit about the age range included in the study, especially if those <5yo are included.

Response: We agree that clarifying the age range is important. We have included this information in both the Methods and Results sections:

“The study was conducted in a routine clinical setting at a tertiary hospital in Madrid, focusing on patients of all age groups who underwent QFT testing. Children were not excluded, as previous research suggests that the QFT assay is accurate in pediatric practice, with good sensitivity [7]”

“The study population comprised individuals aged from 0.05 to 98.52 years, with a mean age of 51.64 years (SD = 19.24). Focusing on younger subgroups, individuals under 18 years of age accounted for 4.7% (466 patients), while those under 5 years represented only 1.2% (122 patients) of the total population.”

Reviewer comment 2. Only 64/758 borderline patients were retested, but the clinical characteristics of this subset are not included in the study nor is the re-testing interval. While not all data may be available, it would be helpful to know if these patients had risk factors or other clinical characteristics different from the overall cohort that prompted the closer follow-up (for example, heavier exposure to TB, country of origin, symptoms of active TB, etc) compared to those that were not re-tested? Additionally, is it confirmed whether these 64 patients are treatment-naive given treatment data was not included as part of the study?

Response: Thank you for your valuable comment. Unfortunately, due to the retrospective nature of the study, we were unable to obtain data on the patients' risk factors, symptoms, or comorbidities. To address the time interval between the initial testing and the repeated testing, we have added the "Months until retesting Mean(SD)" and "Range" columns to Table 2. Additionally, we have included Supplementary Table 1, where the data for continent and mean age can be found in detail. We hope these additions provide greater clarity on the patients' demographics. Moreover, all individual-level data for each patient and result will be made publicly available in a repository.

Reviewer comment 3. While re-testing of borderline QFT results may provide additional diagnostic clarity, retesting ~8% of patients is an additional health care cost and risks losing patients to follow-up, some of whom would go on to develop active TB, but has the advantage of potentially avoiding unnecessary LTBI treatment in a subset of individuals. It may be helpful to include in the discussion potential positive and negative consequences of initiating this intervention.

Reviewer comment 4. It would also be helpful to include in the discussion the effects of serial testing on public health in both low endemicity and high endemicity settings.

Response to comments 3 and 4: Thank you for your valuable contributions. We have added some paragraphs addressing these topics in the discussion section:

“Interpreting diagnostic test results requires careful consideration of the clinical and epidemiological context. The risk of developing active TB varies among individuals with positive IGRA reactions. Studies have demonstrated a correlation between IGRA result magnitude and the incidence of active TB, including cases within the borderline range, suggesting a dose-response relationship [2]. However, in regions with low TB prevalence, individuals presenting with borderline QFT results often experience reversion to negative upon retesting. A study involving 58,539 subjects in a low-endemicity country found that among those with initial borderline results who were retested, 38% reverted to negative, with no cases of incident active TB reported within two years. This finding suggests that a significant proportion of initial borderline results may represent false positives [3]. Consequently, the risk of missing a true LTBI in such cases is relatively low, and avoiding unnecessary treatment minimizes the potential for adverse effects and disruptions to daily life.

Conversely, in populations at higher risk for TB progression, such as immunosuppressed individuals, the emphasis shifts toward maximizing test sensitivity, even at the expense of specificity. Failure to diagnose and treat LTBI in these patients can lead to severe outcomes, including the development of active TB disease. Therefore, integrating other risk factors—such as recent exposure to active TB cases, underlying comorbidities, and the degree of immunosuppression—along with underlying determinants such as access to healthcare, socioeconomic status, and level of vulnerability is essential in guiding clinical decisions [4].

Contrasting evidence suggests that most borderline QFT results reflect true antigen-specific responses rather than random variability, and that the presence of various non-IGRA risk factors and parameters of TB infection showed a gradient along increasing quantitative QFT results [5, 6]. This study provided interesting data on QFT results, but it is important to note that the study was conducted within the context of a contact study, and not in other settings, where the interpretation of the results and the risk of progression to active TB may vary significantly. We agree with Wikell et al. [3] that the inherent variability of the technique should be given due consideration, and that further study is required to ascertain the clinical significance of borderline results in terms of the development of active TB in a range of clinical settings.

Furthermore, given the proportion of borderline QFT results observed in our study and the low incidence of TB progression within this subgroup, one could argue that clinical follow-up alone may suffice, with treatment initiated as necessary. However, refining diagnostic classification strategies would enhance the efficient allocation of healthcare resources and could also reduce the stigma associated with a positive diagnosis. Targeting preventive treatment to individuals most likely to benefit not only optimizes cost-effectiveness but also reduces unnecessary interventions in those at low risk for disease progression.

From a public health perspective, any modifications to TB screening protocols must consider the availability of resources—both material and personnel—to ensure appropriate patient follow-up. Implementation strategies should be tailored to local epidemiological conditions and healthcare infrastructure to maximize feasibility and impact.

Given the significant morbidity and mortality associated with TB, further research is essential to determine which individuals with IGRA conversion or reversion are most likely to benefit from TB preventive treatment, and more studies are warranted across diverse geographical regions and ethnic populations. A deeper understanding of the clinical significance of borderline results for specific patient populations will be crucial in refining LTBI management strategies and improving patient outcomes.”

And we also have added a brief statement in the conclusions of the study:

“While repeat testing offers valuable insights, it may not always be feasible, and further studies are needed to assess the long-term clinical outcomes of borderline QFT results.”

Reviewer #3:

The manuscript entitled "Refining the Diagnostic Approach to Latent Tuberculosis Infection with Quantiferon Gold Plus: A Retrospective Analysis of Borderline Results" needs these revisions

Reviewer comment 1: Please check keywords according to MESH browser; delete QuantiFERON-TB Gold Plus and borderline results from keywords.

Response: Thank you for your valuable comment. We have revised the keywords and have replaced "QuantiFERON-TB Gold Plus" and "borderline results" with more appropriate terms according to the MESH browser.

Reviewer comment 2: Please write about the age range of patients

Response: We agree that clarifying the age range is important. We have added this information in results section:

“The study population comprised individuals aged from 0.05 to 98.52 years, with a mean age of 51.64 years (SD = 19.24). Focusing on younger subgroups, individuals under 18 years of age accounted for 4.7% (466 patients), while those under 5 years represented only 1.2% (122 patients) of the total population.”

Reviewer comment 3: What are the rules for repeat testing, what is the minimum time interval for repeat testing.

Response: Thank you, we agree with your suggestion and have expanded on this topic in the Methods section:

“Patients were tested based on clinical indications, such as suspected latent or active tuberculosis, or for screening purposes. Repeat testing was carried out when deemed necessary by the attending physicians, in consultation with microbiologists. This occurred in cases where there were inconsistencies with expected results, a high clinical suspicion of TB, or indeterminate results.”

To clarify the time intervals for repeat testing, we have included two new columns in Table 2: 'Months until retesting (Mean ± SD)' and 'Range'. This addition provides a clear overview of the minimum time intervals and variability between initial and repeated testing.

Reviewer #4:

The manuscript is well written and highlighted the important subject of diagnosing LTBI, and more importantly to the borderline outcomes interrupting the results which alters in the retesting. Applause to the authors for extracting and analysing such data to develop this manuscript. There are a couple of minor comments/suggestions to this.

Reviewer comment 1: Line 75 - 86: this section could explore further the setup of health facility which the study was conducted, such as TB facility or any other, how patients were selected to test, QTF testing criteria, criteria for retesting of borderline results, and treatment and follow up or further care after LTBI diagnosis.

Response: Thank you for your valuable comment. We agree with your suggestion and have expanded on this topic in the Methods section. The following additional information has been included:

“A retrospective review of patients with QFT results between 2019 and 2023 was conducted, including demographic data such as age, gender, and country of birth. These variables have been associated, along with borderline test results, with both the likelihood of conversions and reversions of Interferon γ Release Assays [1]. The study was conducted in a routine clinical setting at a tertiary hospital in Madrid, focusing on patients of all age groups who underwent QFT testing. Children were not

---

## [Decision Letter · Decision Letter 1]

31 Jul 2025

Refining the Diagnostic Approach to Latent Tuberculosis Infection with Quantiferon Gold Plus: A Retrospective Analysis of Borderline Results

PONE-D-24-52042R1

Dear Dr. Roa,

We’re pleased to inform you that your manuscript has been judged scientifically suitable for publication and will be formally accepted for publication once it meets all outstanding technical requirements.

Kind regards,

Frederick Quinn

Academic Editor

PLOS ONE

Additional Editor Comments (optional):

Reviewers' comments:

Reviewer's Responses to Questions

**Comments to the Author**

1. If the authors have adequately addressed your comments raised in a previous round of review and you feel that this manuscript is now acceptable for publication, you may indicate that here to bypass the “Comments to the Author” section, enter your conflict of interest statement in the “Confidential to Editor” section, and submit your "Accept" recommendation.

Reviewer #4: All comments have been addressed

Reviewer #5: All comments have been addressed

Reviewer #6: All comments have been addressed

2. Is the manuscript technically sound, and do the data support the conclusions?

Reviewer #4: Yes

Reviewer #5: Yes

Reviewer #6: Yes

3. Has the statistical analysis been performed appropriately and rigorously? 

Reviewer #4: Yes

Reviewer #5: Yes

Reviewer #6: Yes

4. Have the authors made all data underlying the findings in their manuscript fully available?

Reviewer #4: Yes

Reviewer #5: Yes

Reviewer #6: Yes

5. Is the manuscript presented in an intelligible fashion and written in standard English?

Reviewer #4: Yes

Reviewer #5: Yes

Reviewer #6: Yes

6. Review Comments to the Author

Reviewer #4: (No Response)

Reviewer #5: Congrajulations to the authors on having successfyllly and adequately addressed all the coments given by the reviewesr. Though uch more could be done with the data set, the present manuscript is well written and complete in itself.. It is an important topic and the study needs to be published as soon as possible

I wonder if you can do a followup paper focussing on only the children below 15 or 18 and on those to 10 years of age and see if there is a difference in the rates of conversion in those two age group. This information will be very helpful in deciding policy regarding preventive therapy.

Reviewer #6: The authors have revised the manuscript based on the reviewers' suggestions, and it is now ready. for publication

7. PLOS authors have the option to publish the peer review history of their article (what does this mean? ). If published, this will include your full peer review and any attached files.

**Do you want your identity to be public for this peer review?** For information about this choice, including consent withdrawal, please see our Privacy Policy .

Reviewer #4: No

Reviewer #5: **Yes: ** Manjula Datta

Reviewer #6: **Yes: ** Paola Di Carlo

---

## [Editor Report · Acceptance letter]

PONE-D-24-52042R1

PLOS ONE

Dear Dr. López-Roa,

I'm pleased to inform you that your manuscript has been deemed suitable for publication in PLOS ONE. Congratulations! Your manuscript is now being handed over to our production team.

Kind regards,

on behalf of

Dr. Frederick Quinn

Academic Editor

PLOS ONE